# Clinical Efficacy and Safety of Lidocaine Tape for Topical Anesthesia of the Oral Mucosa: A Preliminary Controlled Trial

**DOI:** 10.3390/dj11120276

**Published:** 2023-11-29

**Authors:** Ryouji Tani, Sachiko Yamasaki, Atsuko Hamada, Mirai Higaki, Yasuyuki Asada, Souichi Yanamoto

**Affiliations:** Department of Oral Oncology, Graduate School of Biomedical and Health Sciences, Hiroshima University, 1-2-3 Kasumi, Minami-ku, Hiroshima 734-8553, Japan; sayamasaki@hiroshima-u.ac.jp (S.Y.); hamaco@hiroshima-u.ac.jp (A.H.); mirai-higaki@hiroshima-u.ac.jp (M.H.); y0520a@hiroshima-u.ac.jp (Y.A.); syana@hiroshima-u.ac.jp (S.Y.)

**Keywords:** administration, topical, infiltration anesthesia, mouth mucosa

## Abstract

Local anesthesia is administered to reduce pain-induced stress during dental treatment. However, local anesthetic injections are extremely painful; thus, methods to minimize this pain should be developed. Clinical studies on the pain-relieving effects of dental topical anesthetics have shown that few topical anesthetics provide fast and adequate pain relief without harming the oral mucosa. We examined the efficacy and safety of lidocaine tape, which has a potent topical anesthetic effect. Lidocaine tape was applied to the oral mucosa of 14 healthy participants, and its suppression effect was assessed by examining the pain intensity at the non-lidocaine tape-applied site using the visual analog evaluation scale and the verbal evaluation scale. Lidocaine tape application significantly reduced visual analog scale (VAS) scores during mucosal puncture compared to non-application (*p* < 0.01). Moreover, lidocaine tape application significantly reduced VAS scores during local anesthetic injection compared to non-application (*p* < 0.001). Adverse events were evaluated using the Common Terminology Criteria for Adverse Events, version 5.0. No adverse events attributed to the application of lidocaine tape were observed in any participant. The findings in this study suggest that the application of lidocaine tape before infiltration anesthesia can reduce patient distress.

## 1. Introduction

Most dental procedures are painful and are therefore performed after infiltration anesthesia is applied through the oral mucosa for localized pain reduction. During infiltration anesthesia, a sharp needle is inserted into the oral mucosa, where the sensory nerves are extremely sensitive, and some patients may experience intense pain, fear, and anxiety. In addition, the patient may feel sick or go into neurogenic shock, making it difficult to continue with dental treatment. Hence, the development of topical anesthetics that can minimize the pain is needed. In addition, providing an initial treatment that is painless is crucial for the continuation of treatment, particularly in individuals with mental retardation and infants requiring multiple dental treatments. Furthermore, with an aging population, the number of older people with cardiovascular diseases is expected to rapidly increase in the future. Since older people with cardiovascular disease have a lower tolerance for stress, pain stimuli during dental treatment may cause changes in the circulatory dynamics, leading to various complications such as arrhythmias and angina pectoris. Therefore, we aimed to develop a painless dental treatment that can minimize stress and significant blood pressure fluctuations in dental patients [1,2].

Numerous clinical studies have investigated the use of commercially available topical anesthetics before infiltration anesthesia; however, few products have demonstrated adequate pain-relieving properties [3,4,5,6]. Moreover, comparing the efficacy of topical anesthetics is difficult because conditions such as the patient’s age, type of topical anesthetic, duration of action of the topical anesthetic agent, and anesthetic site vary between studies [7].

Penles^®^ Tape (Maruho, Osaka, Japan) is impregnated with 60% lidocaine and is primarily indicated for pain relief during the insertion of an indwelling needle [8]. The high concentration and quick release of lidocaine result in a powerful topical anesthetic effect; however, its use for pain relief in the oral mucosa has not been reported. Previous studies have evaluated the pain-suppressing effect of lidocaine tape during needle insertion and drug injection for infiltration anesthesia [4,5]. These studies used a tape with a lidocaine concentration of 20%, and no adverse events or other safety issues associated with its use were observed. In this study, a clinical trial was conducted to determine the efficacy and safety of a tape containing 60% lidocaine.

## 2. Materials and Methods

### 2.1. Study Design

This clinical study was conducted in accordance with the Clinical Research Law enacted in April 2018 in Japan and the tenets of the Declaration of Helsinki as revised in 2013. This study was planned as a specified clinical trial according to the guidelines established by the Ministry of Health, Labour and Welfare in Japan. The protocol of this study was approved by the Hiroshima University Certified Review Board (approval number: CRB2022-0017) and registered with jRCT (trial number: jRCTs061230003) on 11 April 2023.

### 2.2. Sample Size Estimation

Fukayama et al. applied 60% lidocaine gel to the oral mucosa of 20 healthy participants and reported that it significantly suppressed pain at the time of needle insertion for infiltration anesthesia compared to that in the control group [6]. We anticipated similar results with the use of a tape containing 60% lidocaine. Therefore, in this study, the mean (standard deviation) of the visual analog scale (VAS) score at the time of puncture was assumed to be 34.8 (30.7) and 0.5 (0.9) for the control and intervention groups, respectively. Assuming a two-tailed significance level of 5%, power of 90%, correlation between groups of 0.3, and normally distributed data, the number of participants required for Wilcoxon’s signed-rank sum test was calculated to be 11. Assuming that a few participants would drop out of this study, the target number of participants was set at 14. 

### 2.3. Eligibility Criteria

Age: ≥18 and <60 years at the time of obtaining consent.Sex: Any.No inflammation or other abnormal findings in the area to which the study drug will be applied (mucosa apical to the maxillary lateral incisor in the vestibular region of the oral cavity).Those who provided free and voluntary written informed consent.

### 2.4. Exclusion Criteria 

Individuals who met any one of the following criteria were excluded from this study.

Individuals who could not make decisions regarding participation in this study.Drug-sensitive individuals who are allergic to any of the drugs used in this study, including Xylocaine dental cartridges.Pregnant women or women who may be pregnant.Those with a relationship with the Department of Oral and Maxillofacial Surgery of our hospital.Other individuals deemed inappropriate as research participants by the principal investigator.

### 2.5. Participants

This study included 14 healthy volunteers. After obtaining informed consent, vital signs (pulse rate, blood pressure, body temperature, and transcutaneous arterial blood oxygen saturation) were measured. The participants had no history of receiving local anesthetics, nerve injury, or allergic reactions to local anesthetics. Investigators used the Research Electronic Data Capture system to input data, confirm that the candidates met the eligibility criteria, and register the participants. Randomization was performed by a lottery method of assigning participants by sex and the order in which they were to receive infiltration anesthesia first in the lidocaine-tape-applied or lidocaine-tape-unapplied area. The person performing the allocation work concealed which group they were allocating the patients to and assigned them to the two groups. Participants were assigned to one of the treatment methods at a ratio of 1:1 according to the Research Electronic Data Capture system. As a result of the allocation, each participant was assigned a group to be taped first on the right side or the left side in the Research Electronic Data Capture system, and the group allocation was recorded.

### 2.6. Interventions

A group; Participants’ vital signs were measured before initiating this study. The labial mucosa of the maxillary anterior teeth was disinfected using a benzalkonium chloride cotton roller. The mucosa of the right maxillary lateral incisor was air-dried, a 10 × 10 mm square of lidocaine tape (Penles^®^ Tape; Maruho, Osaka, Japan; 1.18 mg/cm^2^) was applied to it, and gauze was placed over the tape to prevent moisture from accumulating on it. The lidocaine tape was held in place for 3 min after application. After removing the tape, the oral mucosa was evaluated for the possible occurrence of any adverse events (e.g., mucositis and pain) using the Common Terminology Criteria for Adverse Events, version 5.0 [9], and the vital signs were measured. A 30-G infiltration anesthesia needle was inserted perpendicular to the oral mucosa at the center of the area from which the lidocaine tape was removed. Pain during needle insertion and submucosal injection of 1.0 mL of 2% lidocaine containing 1:80,000 epinephrine (ORA^®^ injectable dental cartridge, GC Showa Pharmaceuticals, Tokyo, Japan) over 1 min was evaluated using the VAS and verbal rating scale (VRS). After 2 min, a 30-G infiltration anesthesia needle was inserted in the mucosal of the left maxillary lateral incisor without applying the lidocaine tape, and 1.0 mL of 2% lidocaine containing 1:80,000 epinephrine was administered submucosally over 1 min. Pain during needle insertion and anesthesia administration was measured using the VAS and VRS. Participants’ vital signs were measured 30 min after removing the lidocaine tape, and the mucosal area to which the lidocaine tape was applied was assessed for any adverse events using the Common Terminology Criteria for Adverse Events, version 5.0. Two different clinicians, instead of the one who administered anesthesia, rated the adverse events from grade 1 to 5.

B group: Participants’ vital signs were measured before study initiation. The labial mucosa of the maxillary anterior teeth was disinfected using a benzalkonium chloride cotton roller. A 30-G infiltration anesthesia needle was inserted perpendicular to the oral mucosa of the right maxillary lateral incisor, and 1.0 mL of 2% lidocaine containing 1:80,000 epinephrine (ORA^®^ injectable dental cartridge, GC Showa Pharmaceuticals, Tokyo, Japan) was administered submucosally over 1 min. Pain during needle insertion and anesthesia administration was evaluated using the VAS and VRS. After 2 min, the mucosa of the left maxillary lateral incisor was air-dried. A 10 × 10 mm square of lidocaine tape was applied to the dried mucosa, and gauze was placed over it to prevent moisture contamination. The lidocaine tape was held in place for 3 min after application. After removing the tape, the oral mucosa was evaluated for possible adverse events (e.g., mucositis and pain) using the Common Terminology Criteria for Adverse Events, version 5.0, and the vital signs were measured. A 30-G infiltration anesthesia needle was inserted perpendicular to the oral mucosa at the center of the area from which the lidocaine tape was removed, and 1.0 mL of 2% lidocaine containing 1:80,000 epinephrine for 1 min. VAS and VRS were used to evaluate pain during needle insertion and anesthesia administration. Participants’ vital signs were measured 30 min after removing the lidocaine tape, and the mucosal area to which the lidocaine tape was applied was assessed for any adverse events using the Common Terminology Criteria for Adverse Events, version 5.0. Two different clinicians, instead of the one who administered anesthesia, rated the adverse events from grade 1 to 5. In this study, the clinician who ultimately evaluated the results was different from the clinician who administered the anesthesia and those who evaluated the adverse events.

### 2.7. Management and Procedures for Research Drugs

Since the research drugs were for unapproved and off-label use only, they were managed and stored in accordance with the established procedures for the management and storage of research drugs. The person in charge of managing the research drugs produced the research drug control chart to record the receipts and disbursements of the research drugs. 

### 2.8. Preparation of Case Reports

The principal investigator prepared a case report form promptly after collecting the specified test and evaluation items. The case report forms were prepared in the Research Electronic Data Capture system, and data were consolidated to exclude information that could identify specific individuals and handled with attention to data reliability. The principal investigator properly maintained the records necessary to ensure the reliability of the data, including source documents.

### 2.9. Specific Methods for Protecting Personal Information

The personal information of the study participants was handled with utmost care. All case reports, documents, and specimens were anonymized by creating a correspondence list for replacing names and descriptions with data unrelated to the research participants. The correspondence list was managed by the principal investigator, stored separately from the anonymized samples and information, and protected with a password. The anonymized samples and documents were stored securely in a locked place, and electronic data were protected with a password. Data management and analysis were performed on a personal computer with adequate security. In addition, any attempt to identify a specific individual from the anonymized samples and information without a rational reason was prohibited. Correspondence lists were not provided to third parties, and no personally identifiable information was disclosed during the reporting or publication of research results.

### 2.10. Endpoints

The primary endpoint was pain suppression during needle insertion, which was assessed by VAS scores. The secondary endpoints included pain suppression during anesthetic injection according to the VAS scores and pain suppression during needle insertion and anesthetic injection according to the VRS scores. 

Safety endpoints included adverse effects (mucositis and pain) on the oral mucosa after lidocaine tape removal which were evaluated using the Common Terminology Criteria for Adverse Events, version 5.0, according to the condition of the mucosa at the application site.

The primary and secondary efficacy endpoints were analyzed using the full analysis set (FAS) and per protocol set (PPS). The FAS included all randomized participants who received treatment at least once, had efficacy data post-randomization, and did not violate the eligibility criteria. The PPS included all FAS participants who completed this study without major protocol violations. The incidence of adverse events was compared between the two groups in the safety analysis set, which included all randomized participants who received the protocol treatment at least once.

### 2.11. Statistical Analysis

Data analyses were performed using JMP Pro software, version 17 (SAS Institute, Cary, NC, USA). The conformity to the normal distribution was evaluated using the Shapiro–Wilk test. Statistical analysis was performed using the Wilcoxon signed-rank sum test, and statistical significance was set at *p* < 0.05.

### 2.12. Monitoring

The principal investigator designated a monitoring officer to ensure that this study was conducted safely and that data were collected accurately in accordance with the research protocol and clinical research methods. The monitoring officer submitted the monitoring results to the principal investigator. The details were in accordance with the monitoring procedures separately stipulated.

### 2.13. Preservation and Handling of Recorded Documents

#### 2.13.1. Storage of Information

Anonymized information obtained during the research period will be stored in the principal investigator’s personal computer until five years have elapsed from the date of completion of the research. After the storage period, the anonymized information will be disposed of in an appropriate manner.

#### 2.13.2. Retention of Documents

The following documents and records will be stored and strictly managed in a lockable storage facility for five years from the date of completion of the research. Electronic data are stored on electromagnetic storage media such as a personal computer or universal serial bus memory device that are independent of the hospital’s local area network or the internet and are protected with a password. When not in use, the storage media are strictly secured in a lockable storage facility. After the storage period is completed, the recorded documents shall be disposed of in accordance with the regulations of the implementing medical institution.

### 2.14. Conflicts of Interest of the Research Organization and Researchers

This study did not receive any financial or other benefits from companies or organizations that are assumed to have an interest in this study, and they did not affect the conduct of this study. Conflicts of interest related to this study were managed in accordance with the standards recommended by the Clinical Research Act. In addition, the lidocaine tape (Penles^®^ Tape; Maruho, Osaka, Japan) used in this study was purchased using research funds from our department.

## 3. Results

### 3.1. Participants’ Characteristics

Fourteen healthy volunteers were enrolled and randomized into one of the two treatment sequences. All participants completed the full study protocol, as shown in the Consolidated Standards of Reporting Trials flow diagram (Figure 1). The median age of the participants was 25.6 years (range, 24–28 years), and seven men and seven women were included in this study.

### 3.2. Outcomes

The Shapiro–Wilk analysis of VAS scores and VRS scores showed that the normal distribution was rejected (*p* < 0.05). The mean VAS scores during puncture at the control and lidocaine-tape application sites were 37.9 ± 16.1 mm and 18.5 ± 14.3 mm, respectively. A significant difference in VAS scores during puncture (*p* < 0.01) was observed between the control and lidocaine-tape application sites. Moreover, the VAS scores during injection of 2% lidocaine with 1:80,000 epinephrine at the control and lidocaine-tape application sites were 27.2 ± 15.2 mm and 6.3 ± 10.3 mm, respectively. A significant difference in VAS scores during injection stimulation (*p* < 0.001) was observed between the control and lidocaine-tape application sites (Figure 2). The mean VRS scores during puncture at the control and lidocaine-tape application sites were 1.14 ± 0.11 mm and 0.71 ± 0.11 mm, respectively. A significant difference in VRS scores (*p* < 0.05) was observed between the control and lidocaine-tape application sites. Furthermore, the VRS scores during injection of 2% lidocaine with 1:80,000 epinephrine at the control and lidocaine-tape application sites were 1.07 ± 0.13 mm and 0.29 ± 0.13 mm, respectively. A significant difference in VRS scores during injection stimulation (*p* < 0.001) was observed between the control and lidocaine-tape application sites (Figure 3). The application of lidocaine tape to the oral mucosa had no significant effect on the vital signs (i.e., systolic and diastolic blood pressure and pulse rate) (Figure 4). No adverse events, such as redness or pain in the mucous membrane, were observed after lidocaine tape removal and similar results were observed 30 min after tape application.

## 4. Discussion

Pain during infiltration anesthesia of the oral mucosa occurs both at the time of needle insertion and during anesthetic injection; however, the pain is reportedly more intense during anesthetic injection than during needle insertion [6]. Although topical anesthetics have been recommended during dental procedures, they are not widely used owing to their limited ability to provide adequate analgesia [7]. Ando et al. examined the pain-suppressing effects of four types of dental topical anesthetic agents in 10 participants using the VAS and reported that topical anesthetic agents did not provide adequate analgesia during injection [10]. Fukayama et al. examined the efficacy of 20% benzocaine gel and 60% lidocaine gel for topical anesthesia and reported that applying 20% benzocaine gel or 60% lidocaine to the gingiva for 20 min for topical anesthesia was not associated with complications such as redness or itching. Topical anesthesia with 20% benzocaine did not significantly reduce the pain caused by mucosal puncture or local anesthetic injection after 20 min when applied using dressing tape; contrarily, 60% lidocaine gel decreased the patient’s perception of pain. The authors reported that the application time of 20 min was too long for actual clinical use and that further research on topical anesthetics is required to shorten the application time [6]. Although Fukayama et al. showed that topical application of lidocaine gel to the oral mucosa can reduce pain at the time of puncture, the optimal application dose has not been investigated [6]. Furthermore, lidocaine gel does not provide adequate anesthesia because the gel flows and does not stay in the area to which it is applied, depending on the region in which it is applied. If the gel spreads to an area where it is not needed, that area also becomes numb, and the saliva dilutes the gel. Furthermore, patient discomfort increases owing to the bitter taste, and controlling the amount of gel applied is difficult; therefore, lidocaine tape is considered more convenient. We believe that the use of lidocaine tape in this study allowed us to control the dosage according to the surface area and duration of application of the tape, and we confirmed a reduction in pain at the time of puncture when lidocaine tape is used. Although lidocaine tape is not indicated for use on the oral mucosa, clarifying the observation items and evaluating them was necessary because of a lack of studies on its safety [4,5]. The surface area of the tape and application time that define the dose were smaller in this study than those used in previous studies. A previous report using DentiPatch showed a lidocaine concentration of 20%, with an area of 2 cm^2^ containing 46.1 mg of lidocaine [4] and the research plan was designed with a focus on safety. Cho et al. reported that the greater the patient’s sense of anxiety, the higher the pain score during local anesthesia; however, the use of a surface anesthetic prevented increased pain perception due to anxiety. In that study, anxiety was not assessed in healthy participants, and the authors emphasized that assessing anxiety in patients with dental disease was necessary in future clinical studies [11]. Intravenous sedation and nitrous oxide inhalation are also used to reduce pain and anxiety in dental patients sedation [12]; however, they are rarely used in general dental clinics as they require special equipment and prolonged treatment time. To date, researchers have attempted to reduce pain during infiltration anesthesia with topical anesthesia using cooling methods [13,14,15,16] and virtual-reality interventions [17,18,19] and by developing motorized infiltration anesthesia machines [20,21,22]. In addition to the aforementioned methods, iontophoresis has been considered a means of avoiding the use of an anesthetic needle during infiltration anesthesia (needle-free anesthesia). DeCou et al. reported on the efficacy of iontophoresis, a needle-free technique in which positively charged lidocaine and epinephrine molecules are drawn into the tissue by an electrical current as an anesthetic for pediatric surgical procedures [23]. Cubayachi et al. investigated the effect of iontophoresis on mucosal penetration of prilocaine hydrochloride (PCL) and lidocaine hydrochloride (LCL), which are predominantly used in dentistry as local anesthetic agents. The authors reported that high concentrations of PCL results in the penetration of the mucosa when iontophoresis is performed with a combination of PCL and LCL at a pH of 7.0, suggesting the possibility of needle-free anesthesia during dental treatment [24]. Do Couto et al. evaluated a mucoadhesive iontophoretic patch to deliver anesthetic to the oral mucosa to achieve needle-free anesthesia [25,26]. Mistry et al. reported that local anesthetic spray agents can be applied deeply using iontophoresis to provide local anesthesia without needle insertion [27]. The disadvantages are the need for special equipment to perform iontophoresis and the time required for the anesthesia to take effect. As society ages, determining ways to safely and reliably reduce pain during local anesthesia in all patients is crucial. 

In this clinical study, we investigated the efficacy and safety of lidocaine tape applied to the oral mucosa to suppress pain during infiltration anesthesia. VAS and VRS were used to measure pain. Accurately assessing the magnitude of pain is difficult, and VAS is the worldwide standard for measuring pain. Although the VAS is generally used as a continuous scale for subjective pain assessment, it is known to have a high degree of interindividual variability. Therefore, VRS was chosen as an additional pain measurement method in our study because it is easy to understand and correlates well with VAS [28]. The application of a tape containing 60% lidocaine significantly reduced pain perception during mucosal puncture and local anesthetic injection compared with that without topical anesthesia. Moreover, none of the participants developed adverse events following the application of lidocaine tape. The tape used in this clinical study contains a high concentration of lidocaine, which can be quickly released into the oral mucosa to which it is applied, thereby providing brief analgesia and reducing the time required for dental anesthesia. In Japan, a tape containing 60% lidocaine was approved for pain relief during needle insertion in 1994, pain relief during molluscum contagiosum removal in 2012, and pain relief during dermal laser radiation therapy in 2013. Post-marketing surveillance studies of the tape containing 60% lidocaine have confirmed its safety and efficacy [29,30].

Conventional topical anesthetics in gel or cream form are diluted by saliva and do not provide adequate anesthesia [6]. Moreover, they may flow into the surrounding oral mucosa and increase discomfort. Adjusting the amount of a topical anesthetic in gel or cream form is difficult once applied. In contrast, lidocaine tape is applied to the puncture site of the infiltration anesthesia needle, facilitating localized surface anesthesia. Nevertheless, because this study was conducted on a small number of young participants, further studies are needed to evaluate the effects of lidocaine-tape application in older patients and patients with systemic diseases. 

This study had some limitations. It has been reported that sex and the order of needle puncture have been reported to affect the perception of pain [7]. Therefore, randomization was conducted using a lottery method to allocate subjects according to sex and the order of needle puncture. Because of the exploratory nature of this study, the planned sample size was small, and the observation interval was short (one day). The limitation of this study is that we analyzed only a small amount of data. The inclusion of so many clinicians with different tasks must also be recognized as a source of bias in the limitations of this study. In the future, placebo-controlled trials with larger sample sizes are needed to thoroughly evaluate the efficacy and safety of the tape containing 60% lidocaine for managing pain during infiltration anesthesia. Furthermore, since this study was conducted in an open-label fashion, bias in the safety assessment had to be considered.

We expect that this product may be approved for insurance coverage and become widely available in the future. Further modifications to the shape of the lidocaine tape (e.g., threads attached to the tape) may be required to prevent accidental ingestion or aspiration.

## 5. Conclusions

Our findings suggest that applying a tape containing 60% lidocaine for 3 min before infiltration anesthesia can safely and significantly decrease pain during mucosal puncture and anesthetic injection, thereby reducing patient stress during dental treatment. The findings in this study can significantly aid in the development of painless dental treatments with minimal stress for dental patients.

## Figures and Tables

**Figure 1 dentistry-11-00276-f001:**
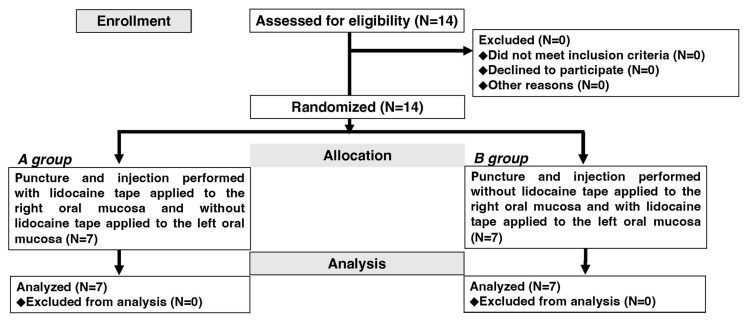
Study flow diagram. Fourteen participants completed the full study protocol (seven participants in each group).

**Figure 2 dentistry-11-00276-f002:**
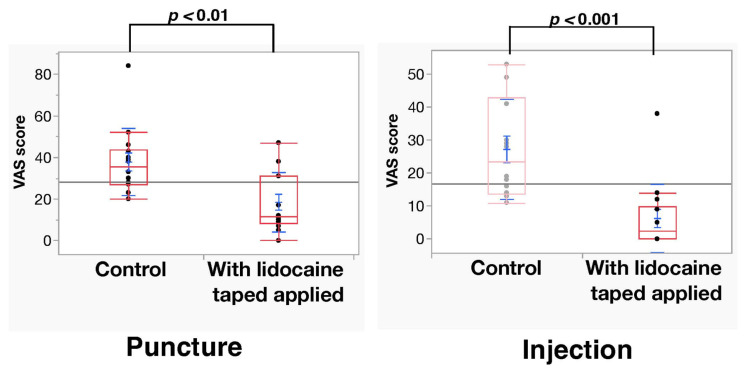
VAS ratings after the two interventions before and after lidocaine tape application (N = 14). VAS, visual analog scale.

**Figure 3 dentistry-11-00276-f003:**
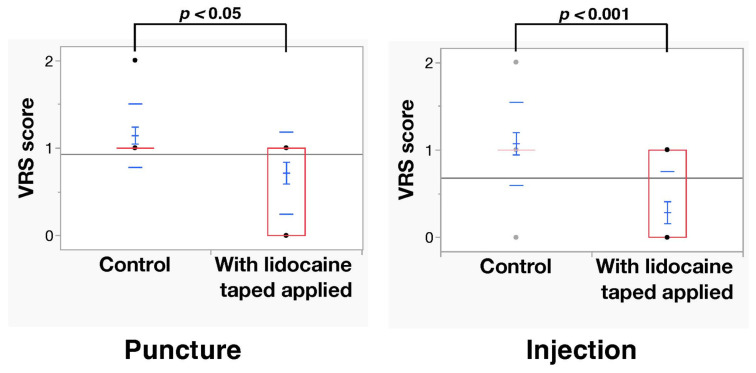
VRS ratings after the two interventions before and after application of the lidocaine tape (N = 14). VRS, verbal rating scale.

**Figure 4 dentistry-11-00276-f004:**
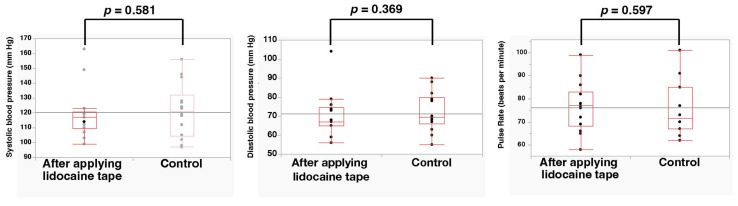
Effect of lidocaine tape application on the vital signs (systolic and diastolic blood pressure and pulse rate).

## Data Availability

The data presented in this study are available on request from the corresponding author.

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
