# Peer review of "Clinical Efficacy and Safety of Lidocaine Tape for Topical Anesthesia of the Oral Mucosa: A Preliminary Controlled Trial"

_dentistry, 2023, doi:10.3390/dj11120276_

Round 1
Reviewer 1 Report
Comments and Suggestions for Authors
Within the scope of my understanding, this is supposed to be a preliminary randomized cross-over clinical trial to assess the effect of an off-label drug use of 60% lidocaine tape compared to none (control) on the clinical efficacy (regarding the pain on insertion and injection of a needle) and safety (regarding adverse effects/events as mucositis and pain or vital signs as blood pressure and pulse rate) on its use for topical anesthesia before infiltration anesthesia in the maxillary lateral incisor region. If this is the case, then a major comment in this study would be that it did not clearly identify and describe this early on in the study e.g. within the title and abstract and throughout the methods section. Throughout the methods section, it was not clear whether the comparison was between the use of lidocaine tape or not using it, or between the use of the tape on the right or left side of the mouth (which, in my opinion, does not have a scientific justification) as shown in the flow diagram presented I the results section!!!
If it were that the comparison was between the use of lidocaine tape or not, then it is a cross-over study with 2 arms/groups (lidocaine tape and control) where the number of patients in each group would be 14 since all the patients (i.e. n=14) which is not in accordance with the presented flow diagram!!
Regarding the title, it does not fully reflect the aim of the study. First, the study assessed clinical efficacy rather than effectiveness since the study was done under controlled conditions rather than real-life conditions. Second, the study assessed safety of the intervention as well which is not mentioned in the title. Third, the study design should be mentioned in the title to facilitate retrieval.
The abstract is insufficiently informative about the study design, conduct and data collection, analysis results and/or interpretation. It does not clearly identify the key information required by the reader e.g. whether there was a control or no? if yes, what exactly was it?!
For the study used for sample size calculation, how come the control show less mean VAS score (pain) than the intervention group?!
If randomization was done, for what exactly was it done? How was it done?! How was allocation concealment done?!
In section “2.6. Interventions”, the mentioned details are confusing and sometimes contradictory requiring extensive rephrasing. It is unclear when the measurement of the vital signs was done in each group? How and when was it done for the control group?! When was it done relative to the application of the lidocaine tape? Was it done after removal of the tape after 3-min application (Line 122-125)?! Or 30 min after removal of the tape (Line 138)?!
What is the rationale of using 2 pain scales (VAS and VRS)?!
Figure legends: Lines 252, 256, “VAS ratings after the two interventions before and after application of the lidocaine tape”. Were vas ratings before and after lidocaine tape application? OR on the intervention side and control side?!
Figure legends: Lines 260: It is unclear when the measurement of the vital signs was done in each group? How and when was it done for the control group?! When was it done relative to the application of the lidocaine tape?
Discussion: the section mostly discusses justification of the aim rather than discuss the methodology used or interpretation of the results or identifying other limitations e.g. study design, randomization method, lack of blinding…?!
Discussion: Line 351 “…..observation interval was short (one day)….” observation duration in this study was only 30 min..!! Any comment?!
Reviewer 2 Report
Comments and Suggestions for Authors
“Clinical effectiveness of lidocaine tape for topical anesthesia of the oral mucosa” was submitted to Dentistry Journal.
This study seeks to examine the pain-suppressing effect of lidocaine tape during needle penetration and local anesthetic injection, utilizing the visual analog scale and verbal rating scales. The authors have concluded that the application of lidocaine tape significantly mitigated pain during mucosal puncture and local anesthetic injection, with no reported adverse events.
The manuscript deals with an interesting issue; however, there are several concerns related to the study.
Title: Please indicate the type of study.
Abstract
Lines 14-16. This information is repeated in the abstract. Please review.
Please provide clear details about the experimental group and the control group.
Present the most important values described in the outcomes section of the main text (line 232).
Keywords: Terms should be reviewed. Please make sure they are MeSH terms.
Introduction
In the first paragraph, bibliographic references should be included as appropriate.
Line 45. Ten references are described without detailing their results. It is recommended to reduce their number and describe the results of the most important ones.
Lines 46-48. Please describe some of those studies and add their references.
Lines 56-59. Describe this sentence with a focus on the objectives.
All the references described in the introduction are over 20 years old. Please use more up-to-date information.
Methods
Lines 61-73. Please significantly condense this paragraph, focusing on the essential information.
Line 105. Age data should be presented in the results section.
In the intervention section, clearly describe the experimental group and the control group.
Line 168. First, describe the primary and secondary outcome variables.
It is mentioned that randomization was performed, but the process was not described. Please review and provide a detailed description of the randomization process.
Absolutely nothing is described about the examiners/operators. Describe the calibration process and its statistical result. Were the clinicians who administered the anesthetic the same as those who assessed the outcomes? Comment on this.
A blinded examiner to the intervention would have reduced assessment biases. Please comment on this in the discussion section.
The statistical analysis is described in a very concise manner. Include the statistical test used to assess the normal distribution of the data, as well as the tests used in the univariate and bivariate analysis. Please review and provide a detailed description.
Results
Table 1 is not necessary as the information has already been described in the text.
The size of Figures 2, 3, and 4 should be increased for better visibility.
Discussion
Lines 263-267. This information has already been provided in the introduction.
Line 270. Please use a more recent reference.
Lines 270-272. Please provide a reference.
Lines 285-293. Please provide references.
Line 296. Describe the doses used in the other studies.
Line 328. Various studies and systematic reviews have described the limitations of VAS. Please describe them.
Lines 342-345. Please provide references.
This study presents more limitations that must be recognized.
As in the introduction, some references are quite old. Please cite more up-to-date bibliographic references.
Comments on the Quality of English Language
Moderate editing
Reviewer 3 Report
Comments and Suggestions for Authors
Well done study. Innovative. Few subjects. Could a shorter version be published as a letter to editor. Is tape commercially available?
Round 2
Reviewer 1 Report
Comments and Suggestions for Authors
Thank you for the significant modifications in the manuscript.
To me, however, there is still a comment regarding the clarity of the randomization and the grouping. Groups should be based on the Intervention (Lidocaine tape) and control (No lidocaine tape) and the flow diagram should reflect that. If the randomization would seem confusing to the reader, then in this case the study design could be designated as a “preliminary controlled clinical trial” rather than a “preliminary randomized clinical trial” and adjust the flow diagram accordingly.
Kindly also include supporting references to the added paragraph in the discussion section regarding the use of VAS and VRS.
Reviewer 2 Report
Comments and Suggestions for Authors
In this revised version the authors have improved the manuscript; However, some points still need to be corrected.
1. As this is a clinical trial with controlled conditions, the authors are evaluating efficacy and non-effectiveness. Please correct this throughout the manuscript including the title.
2. The authors indicate that two clinicians evaluated the adverse events. Therefore, the calibration protocol and the statistical test with its result that corroborates its agreement should be included in the manuscript.
3. For greater clarity, the text should include that the clinician who administered the anesthesia and the clinician who evaluated the results were different. Here another question arises related to point 2. Is this clinician who evaluated the results also different from those who evaluated the adverse events? All this must be clarified.
4. The inclusion of so many clinicians with different tasks must also be recognized as a source of bias in the limitations of the study.
5. The normal distribution of quantitative data should always be evaluated even in small samples (Shapiro-Wilk). This shortcoming must also be recognized in the limitations of the study.
Comments on the Quality of English Languageminor
Round 3
Reviewer 2 Report
Comments and Suggestions for Authors
Manuscript ID dentistry-2668409
Title: Clinical effectiveness and safety of lidocaine tape for topical anesthesia of the oral mucosa: A randomized controlled study
Comment 1: As this is a clinical trial with controlled conditions, the authors are evaluating efficacy and non-effectiveness. Please correct this throughout the manuscript including the title.
Authors response: As you indicated, this is a clinical trial under controlled conditions; hence, the title has been revised to "Clinical effectiveness…”
Additional comment to this answer: efficacy is about how well an intervention works under ideal and controlled conditions, often in a clinical trial setting, whereas effectiveness assesses how well the intervention performs in the real-world clinical practice, considering the broader and more diverse patient population encountered in routine healthcare.
Remember to correct this observation throughout the entire manuscript.
Comment 4: The inclusion of so many clinicians with different tasks must also be recognized as a source of bias in the limitations of the study.
Authors response 4: Thank you for your review and valuable suggestion. As you mentioned, the roles of the study were shared by several researchers, and we recognized that this could be a source of bias. However, we did not switch roles throughout the study; hence, we believe that the 14 participants were evaluated equally.
Additional comment to this answer:
· Different clinicians may interpret data differently, leading to variability in their assessments. This can introduce subjectivity and decrease the reliability of the results.
· Clinicians may apply different standards or criteria when evaluating outcomes, especially if there is a lack of standardized guidelines. This can result in inconsistent and biased assessments.
· If clinicians are not blinded to treatment assignments, their knowledge of the assigned interventions may introduce bias into the evaluation process.
· Clinicians need to be adequately trained to ensure consistency in their evaluations. Without proper training and standardization, there is a risk of misinterpretation and inconsistency in the assessment process.
Considering the foregoing, this limitation should be acknowledged.
Comment 5: The normal distribution of quantitative data should always be evaluated even in small samples (Shapiro-Wilk). This shortcoming must also be recognized in the limitations of the study.
Authors response 5: Thank you for your review and valuable suggestion. As you indicated, we have performed a Shapiro–Wilk analysis of the normal distribution of VAS and VRS and confirmed that the normal distribution was rejected. We are aware that this is a limitation of the study since we analyzed only a small amount of data.
Additional comment to this answer:
This statistical information does not appear in the respective section.
Comments on the Quality of English Language
minor
Author Response
"Please see the attachment.
